# Real-Life Effectiveness of Mepolizumab in Refractory Chronic Rhinosinusitis with Nasal Polyps

**DOI:** 10.3390/biomedicines11020485

**Published:** 2023-02-08

**Authors:** María Sandra Domínguez-Sosa, María Soledad Cabrera-Ramírez, Miriam del Carmen Marrero-Ramos, Delia Dávila-Quintana, Carlos Cabrera-López, Teresa Carrillo-Díaz, Jesús Javier Benítez del Rosario

**Affiliations:** 1Department of Otorhinolaryngology, Hospital Universitario de Gran Canaria Doctor Negrín, 35010 Las Palmas de Gran Canaria, Spain; 2Department of Quantitative Methods in Economics and Management, Universidad de las Palmas de Gran Canaria, 35017 Las Palmas de Gran Canaria, Spain; 3Department of Pneumology, Hospital Universitario de Gran Canaria Doctor Negrín, 35010 Las Palmas de Gran Canaria, Spain; 4Department of Allergology, Hospital Universitario de Gran Canaria Doctor Negrín, 35010 Las Palmas de Gran Canaria, Spain

**Keywords:** chronic rhinosinusitis, nasal polyps, mepolizumab, aspirin exacerbated respiratory disease, SNOT-22, nasal polyp score

## Abstract

The aim of this study was to evaluate the efficacy of mepolizumab in patients affected by chronic rhinosinusitis with nasal polyps (CRSwNP) in real-life. A single-center retrospective observational study was conducted on severe CRSwNP patients treated with mepolizumab. Nasal endoscopic polyp score (NPS), visual analogue scale (VAS) symptom score, sinonasal outcome test (SNOT-22), asthma control test (ACT) score, fractional exhaled nitric oxide (FeNO), eosinophils blood cells and prednisone intake were assessed at baseline and after 6 months. A total of 55 patients were included; 49 patients (89%) presented with asthma; aspirin exacerbated respiratory disease (AERD) in 28 patients (51%). A statistically significant decrease in the SNOT-22 score was observed (median difference −63; 95% CI: −68; −58; *p* < 0.001) with median t0 76 and IQR (61;90) to t6 10 (5;15). A reduction in NPS, median t0 NPS 4; (IQR:4;6), median t6 NPS 1; (IQR:0;1) *p* < 0.001, was greater in patients with AERD. The median baseline VAS score was 6 (IQR:6;7) and the differences between t0 and t6 were statistically significant *p* < 0.001. Significant changes in blood eosinophils cells, median t0 500 cell/mcl (IQR:340;830), median t6 97 cell/mcl (IQR:60;160) *p* < 0.001, were greater in patients with AERD. Mepolizumab treatment effects have been demonstrated with significantly reduced symptoms, polyp scores, blood eosinophils and systemic corticosteroid use, resulting in an increased health-related quality of life in patients with severe CRSwNP, regardless of the presence or absence of asthma or AERD.

## 1. Introduction

Chronic rhinosinusitis with nasal polyps (CRSwNP) is a chronic inflammatory disease of the paranasal sinuses, and an important subtype phenotype of chronic rhinosinusitis (CRS) [1,2]. It has an estimated prevalence of 4.2% in the general population in the USA and 4.3% in Europe [3,4]. Clinically, CRS is defined by the presence of two or more symptoms, one of which should be nasal blockage, loss of smell, facial pressure, or nasal discharge, for more than 12 weeks and supported by signs of sinus inflammation detected by imaging procedure and/or nasal endoscopy [1,5]. It is phenotypically classified by the presence or absence of nasal polyps into CRSwNP and chronic rhinosinusitis without nasal polyps (CRSsNP) [6]. CRSwNP is a heterogeneous disorder that may be associated with other systemic disorders [7]. CRS can also have a substantial impact on health-related quality of life (HRQoL) [8] and the phenotype CRSwNP is a debilitating disease that has a significant annual economic burden [9]. CRSwNP has been associated with type 2 inflammation involving eosinophil infiltration and cytokine production (IL-4, IL-5 and IL-13) with IL-5 playing a key pathogenic role [7,10,11,12,13]. The standard of care for CRSwNP includes intranasal and short courses of oral corticosteroids (OCS) as well as nasal surgery [1]. OCS reduce inflammation, nasal polyp size and improve symptoms; however, potential side effects are associated with their long-term use [1,14]. Many of the patients with CRSwNP who failed the standard of care need endoscopic sinus surgery (ESS) but recurrence of nasal polyps after nasal surgery is also common. Rates are estimated at 40% to 80% at 3 to 12 years after surgery [15,16,17]. Therefore, new treatment options are necessary. With the introduction of precision medicine, a new era has rapidly developed in airway inflammatory diseases [18,19] such as cystic fibrosis, asthma, CRS and rhinitis. The co-incidence of CRSwNP and asthma is about 20–70% [3,12,20]. Previous studies show that up to 15% of patients with CRSwNP have comorbid asthma and an intolerance to inhibitors of cyclooxygenase 1 (COX-1) [21,22]. For Stevens WW et al. [22] the term aspirin exacerbated respiratory disease (AERD) specifically refers to patients with CRSwNP and asthma who also develop respiratory reactions to COX-1 inhibitors and is the most severe sub-phenotype of CRSwNP [22,23]. Biological therapy goals consist of improving control in those patients who would generally need repeated high-dose and/or long-term OCS or even revision surgery [24]. Current biologic therapies have become an option for the treatment of refractory CRSwNP, associated with asthma, severe disease and recurrence after surgery, targeting the type 2 inflammation. Mepolizumab is a humanized monoclonal antibody (mAb) against IL-5, a cytokine that promotes eosinophil development and survival. Administration of a dose of 100 mg of mepolizumab subcutaneously every 4 weeks has been approved since 2015 for the add-on maintenance treatment of adults with severe eosinophilic asthma (SEA) [25]. Phase 3 trials demonstrated subjective and objective efficacy and FDA-approved mepolizumab for CRSwNP in July 2021 [26]. Therefore, mepolizumab had not been approved for use in CRSwNP during the period of our study. The choice of mepolizumab as biologic therapy was made by pneumologists and allergists from our health area. The aim of this study was to evaluate the efficacy of mepolizumab on sino-nasal aspects in the treatment of uncontrolled CRSwNP, with or without asthma in a real-life setting over the first six months of treatment.

## 2. Materials and Methods

We conducted a single-center observational study in a real-life setting using data collected from 55 patients with CRSwNP, which were treated with 100 mg of mepolizumab, administered subcutaneously once every 4 weeks, in addition to the standard of care from January 2018 until January 2022 at our healthy area.

According to the European Position Paper on Rhinosinusitis and Nasal Polyps (EPOS) 2020, standard treatment throughout the study period consisted of daily intranasal corticosteroids, saline nasal irrigations and OCS or antibiotics, or both, as needed [1].

The hospital ethical committee approved the study protocol. An informed consent form was obtained from all patients before data collection. The diagnosis of CRSwNP was confirmed in line with EPOS 2020 criteria [1].

The medical records of 55 patients were reviewed and analyzed for preoperative demographic characteristics and medical history, including smoking history, comorbid asthma, AERD, allergic rhinitis and atopy.

Asthma was diagnosed according to the 2022 Global Initiative for Asthma’s definition [27] and allergic rhinitis was diagnosed according to the 2019 Allergic Rhinitis and its Impact on Asthma [28]. All patients with severe uncontrolled asthma met the criteria according to the ATS/ERS guidelines [29].

The following outcomes were assessed at baseline (t0) and after 6 months (t6): Nasal endoscopic polyp score (NPS), visual analogue scale (VAS) symptom score, the validated sinonasal outcome test (SNOT-22), asthma control test (ACT) questionnaire, fractional exhaled nitric oxide (FeNO) and prednisone intake. Sinonasal Computed Tomography (CT) scans were performed at baseline. These CT scans were scored according to the Lund–Mackay staging system [30].

The polyp score was assessed at each study visit by trained ear, nose and throat staff. The nasal polyp size at each side of the nasal cavity was measured endoscopically by using the NPS: 0–4, a higher score representing a larger size of polyp. NPS was assessed for each nostril using the sum of both unilateral scores. Each nostril score ranges from 0 (no polyps) to 4 (large polyps causing almost complete obstruction of the inferior meatus) giving a maximum score of 8 [31].

At baseline and the 6-month follow-up visit, patients recorded symptoms (nasal obstruction, nasal discharge, postnasal discharge, olfactory disorders, facial pain and overall symptom scores) using VAS scale (0–10); scores were divided by 10 and reported across a range from 0 (none) to 10 (as bad as you can imagine), according to their current state. VAS score was used as in previous studies of nasal polyps [26,32]. We reported VAS score for each symptom and a mean composite overall VAS symptom score (combining scores for nasal obstruction, nasal discharge, throat mucus, facial pain and loss of smell).

The 22-item SNOT-22, a nasosinusal disease-specific measure of HRQoL was used. It is a questionnaire constituted of 22 CRS-related items scored from 0 to 5 (total score ranges from 0 to 110 with higher scores indicating worsening of symptoms). It assesses the severity of symptoms reported by patients due to the disease and was completed by patients at baseline (t0) and 6 months (t6) [33].

We used the ACT, a self-administered questionnaire and the FeNO, that was measured using an electrochemical analyzer (FeNO: NIOX; MINO handheld, Aerocrine AB, Solna, Sweden), for assessing asthma control.

Blood samples were collected for hematology in peripheral blood for the determination of total serum IgE at baseline and the determination of eosinophil cells at baseline and week 24. Safety (review of adverse events and serious adverse events) were assessed at each visit. Biopsies of nasal polyps were performed in all patients who underwent ESS. We did not perform CT scans after treatment to avoid unnecessary ionizing radiation.

Intake of prednisone and exacerbations (calculated as episodes requiring OCS treatment for at least 3 days or a single intramuscular treatment), were also included in the database for all time points. Treatment compliance was strictly assessed at each clinical visit.

## 3. Statistical Analysis

Categorical variables were summarized using frequencies (n) and percentages, while continuous variables were summarized through the mean and standard deviation (SD) in case of normality, or with the median and interquartile range (IQR: 25th–75th percentile) in case of non-normal distributions assessed by a Kolmogorov–Smirnov test. The percentages were compared, as appropriate, using the Chi-square (χ^2^) test, the exact Fisher test or the McNemar test. The comparisons between the baseline and final values of outcomes were carried out using the Wilcoxon signed-rank test for paired data. Non-parametric point estimates and 95% confidence intervals for treatment effect (median difference) were constructed using the Hodges–Lehmann method [34]. The Mann–Whitney test was used to compare the distribution between two groups. Statistical significance was set at *p* < 0.05. Data were analyzed using IBM SPSS Statistics, v.27.0 software.

## 4. Inclusion and Exclusion Criteria

In our study, we included patients older than 18 years with symptoms of severe CRSwNP, defined as follows:

Nasal obstruction VAS score > 5;

Global symptom VAS score > 5;

Patients with CRSwNP and asthma;

Non asthma but CRSwNP and multiple previous nasosinusal surgical procedures;

Endoscopic NPS ≥ 4, with a minimum score of 2 in each nasal cavity;

≥1 polyp surgeries in the last 10 years;

Maintenance treatment with intranasal corticosteroids;

Presence of at least 2 of the following symptoms, for ≥12 weeks: nasal obstruction/blockage/congestion or nasal discharge (anterior or posterior nasal drip) and at least ≥1 of the following symptoms: nasal discharge, facial pressure, or pain, decreased or lost sense of smell;

Patients who met the following criteria were excluded from our study:

Certain comorbid disorders and other medical conditions, such as cystic fibrosis, Young syndrome nasosinusal, Kartagener syndrome;

Acute sinusitis or upper respiratory tract infection within 2 weeks prior to inclusion;

Asthma exacerbation requiring hospitalization within 4 weeks prior to inclusion;

Contraindication for nasal surgery, such as refusal by the patient, contraindication for general or local anesthesia;

Biological or immunosuppressive treatment before inclusion.

## 5. Results

We analyzed the data from 55 patients with CRSwNP, who were treated with 100 mg of mepolizumab, administered subcutaneously once every 4 weeks for 24 weeks, from January 2018 to January 2022 at our center. Our results show a female predominant sex (64% women and 36% men), in a ratio of 2:1. There were no apparent differences in the mean age and gender of the patients, with a mean age of 53.7 years and a standard deviation (SD) of 11.0. Patients’ socio-demographic characteristics, including smoking status and comorbid conditions, are displayed in Table 1.

At baseline 49 patients (89%) presented with asthma; 44 patients (80%) presented with uncontrolled asthma and were on continuous OCS therapy; and 5 patients (9%) presented with partly controlled asthma. AERD was described in 28 patients (51%) and 29 patients (53%) presented with allergies. Of the six non-asthmatic patients, five had to undergo repeated ESS, before starting mepolizumab treatment with a mean of 3.6 interventions per patient. Among all patients, nine (16%) had a history of smoking tobacco but were not active smokers at the time of the study.

Nasal polyps commonly present clinically with nasal obstruction and a reduction/loss of smell [35]. Almost all our patients had nasal obstruction (94.5%) and hyposmia/anosmia (94.5%) at baseline (t0), and 53 patients, that is, 96.4% of the total, received treatment with OCS before mepolizumab and only 2 (3.6%) continued with doses of OCS after treatment with mepolizumab. Of these two patients, both were asthmatics (100%), one of them underwent recurrent surgery and the use of OCS was due to poor asthma control.

All patients underwent ESS, with the median number of surgeries per patient and the IQR being 1 (0;2); (with a range from 0 to 8). A total of 20 patients (36.4%) had two or more previous nasal surgeries, 30 surgeries were performed in total, 19 in women and 11 in men. It should be noted that for all repeated surgeries underwent before starting mepolizumab treatment; to date, we have not carried out any ESS on any patient treated with mepolizumab.

The summary of pre-treatment and post-treatment outcomes of interest is presented in Table 2. In these 55 patients, at baseline (t0), the median SNOT-22 score was 76 (IQR:61;90), and after treatment with mepolizumab, patients experienced a statistically significant decrease (improvement) in the SNOT-22 score, with a median difference of -63 points (95% CI: −68; −58) *p* < 0.001 at t6 (Figure 1) [32,36]. HRQoL evaluated by SNOT-22 score in patients treated with mepolizumab improves with high certainty, that is, had a clinically meaningful improvement (change in score ≥ MCID set to 8.9) [37].

The median baseline (t0) ACT score was 11 (5;25) and after 24 weeks of mepolizumab (t6) 21 (8;25). A Wilcoxon signed-rank test showed an improvement in terms of median ACT score *p* < 0.05.

A decreasing trend was measured for FeNO, median baseline t0 was 50 (36;130) and t6 23 (14;36). A Wilcoxon signed-rank test showed an improvement in terms of median FeNO score that was statistically significant (*p* < 0.05).

The VAS score could range from 0 to 10, as used in previous studies of nasal polyps [30]. The median baseline VAS score was 6 (IQR:6;7) and the differences between scores at baseline and posttreatment were statistically significant at Wilcoxon signed-rank test with *p* < 0.001. The corresponding Hodges–Lehmann estimate for difference in medians was −4 (95% CI: −4; −4) (Figure 2). Overall, there was no significant difference in VAS score between patients with AERD (median difference: −4) and those without AERD (median difference: −4).

There was a statistically significant decrease in NPS before treatment (median: 4; IQR:4;6) and after treatment (median: 1; IQR:0;1) with a median difference of −4 and a *p* < 0.001 (Figure 3a and Table 2). The reduction in NPS was greater in patients with AERD (Figure 3b). In AERD patients, NPS score decreased by 4.5 points (95% CI: −5; −3.5) *p* < 0.001. In contrast, patients without AERD had a smaller median decrease (median difference: −4; 95% CI: −4.5; −3) *p* < 0.001.

A Wilcoxon signed-rank test showed that the 24-week mepolizumab treatment elicited a statistically significant change (median difference: −460; 95% CI: −610; −353) in patient eosinophils cells with a median baseline of 500 cell/mcl (IQR 340;830). In fact, the median number of eosinophils was 97 cell/mcl (IQR 60;160) post-treatment *p* < 0.001 as shown in Figure 4a. We found a greater reduction in the level of eosinophils among patients with AERD compared to non-AERD, while there are no differences based on the existence of previous surgeries (Figure 4b).

For all clinical parameters, a statistically significant difference is found at t0 and t6 and, hence, it can be attributed to the effect of the mepolizumab treatment: SNOT-22 score (*p* < 0.001), NP score (*p* < 0.001), blood eosinophil count (*p* < 0.001), FeNO (*p* < 0.05), NPS score (*p* < 0.001) and blood eosinophil count (*p* < 0.001). Statistically significant differences were also found for individual sinonasal symptoms before and after the initiation of mepolizumab therapy such as nasal obstruction (*p* < 0.001), hyposmia/anosmia (*p* < 0.001) or even OCS use (*p* < 0.001).

Improvements in loss of smell, which is one of the most bothersome symptoms for patients, were greater in patients with fewer previous surgeries while no significant differences were found at Mann–Whitney (*p* = 0.648) between AERD and non AERD patients (Figure 5).

All patients performed a sinus CT scan before the treatment and the median Lund–Mackay score was 20 (IQR:16;22) but we did not perform CT scans after treatment. Differences between CT scan scores at AERD patients and non-AERD were significant for the Mann–Whitney test with *p* = 0.024 (95% CI: −4;0).

## 6. Discussion

This study evaluated the efficacy and safety of mepolizumab in the treatment of patients with recurrent, refractory severe bilateral CRwNP, regardless of the association with other comorbidities, such as the presence of asthma, AERD and allergic rhinitis. Since all patients had undergone one or more nasal surgeries prior to study entry, their CRSwNP was considered refractory to medical and surgical treatment [26].

Biologic therapies are a good alternative to patients with severe CRSwNP, which is associated with asthma comorbidity, more severe disease and recurrence after surgery. In recent years, several mAbs have been approved to treat other conditions associated with type 2 immune responses, including CRSwNP, in addition to severe asthma, with minimal side effects [38].

The use of anti-IL-5 biologic therapies for the treatment of SEA revealed a significant improvement in sinonasal symptoms in patients with CRwNP and raised expectations about the possible benefit of these drugs in the treatment of patients with difficult-to-control type 2 nasal eosinophilic polyposis [25].

Mepolizumab is a humanized mAb against IL-5, a cytokine that promotes eosinophil recruitment, development and survival and prevents its engagement with IL-5 receptor alpha (IL-5R𝛼) [39]. Mepolizumab does not seem to interfere with other cytokines due to its high-affinity interaction and presents a good tolerance and safety profile [40].

Administration of a dose of 100 mg of mepolizumab subcutaneously every 4 weeks has been approved since 2015 for the add-on maintenance treatment of adults with SEA and at a dose of 300 mg for eosinophilic granulomatosis with polyangiitis or hypereosinophilic syndrome in the USA [41,42].

Asthma comorbidity and intolerance to inhibitors of cyclooxygenase 1 (COX-1) is observed in approximately 15% of patients with CRSwNP. Patients with AERD tend to have a more severe form of the disease, with greater sinonasal inflammation, their polyps tend to grow faster and they require a greater number of sinus surgeries due to the recalcitrant nature of their disease [21,22,23]. Our data suggest that mepolizumab is efficacious in patients with or without these comorbidities, and the safety profiles are similar in these patients despite the high disease burden. Among patients with or without AERD comorbidity, we did not find any statistically significant difference in any of the analyzed clinical parameters, except for eosinophilia levels and NPS. AERD showed a more significant reduction in blood eosinophilia and NPS than patients without AERD.

The negative impact on CRSwNP patients’ HRQoL is evident, in mental and physical health, social and emotional function, subjective and objective sleep dysfunction, as well as work absenteeism conditioned by nasal symptoms (rhinorrhea, nasal congestion, loss of smell) and symptoms derived from lower airway inflammation [43].

A reduction in nasal symptoms has been described in real-life studies (RLSs) [44,45]. We found statistically significant differences for nasosinusal symptoms such as nasal obstruction and loss of smell before and after the initiation of mepolizumab therapy. Additionally, mepolizumab reduced the use of OCS and improved nasal symptoms including loss of smell which is one of the most debilitating and bothersome symptoms, in patients with severe, bilateral CRSwNP.

In our study, improvements in loss of smell were greater in patients with fewer previous surgeries, which has also been observed in the SYNAPSE study [26], while no significant differences were found between AERD and non AERD patients. Multiple ESS can reduce the sense of smell [46]. Therefore, the history of repeated nasal surgeries probably contributed to the improvements seen in loss of smell VAS scores. This finding emphasizes the need for treatments that reduce repeat surgeries [26].

ESS plays an important role in patients with severe nasal polyposis despite appropriate medical treatment [1]. Surgical approaches can range from a simple polypectomy to complete removal of polyp and sinus mucosa from all sinuses involved. However, the recurrence rate following ESS is high and very impactful for patients, and repeated surgeries are often required. Mepolizumab treatment reduces the need for surgery and decreases the burden associated with surgery for patients with nasal polyposis [32,47].

We report that none of our patients receiving mepolizumab underwent actual revision nasal surgery. The reduction in nasal surgery achieved by treatment with mepolizumab is in line with the European Position Paper on Rhinosinusitis and Nasal Polyps 2020 recommendations to reduce surgical procedures [1]. The risks associated with surgery reported by patients as well as its disruptive nature should be factors considered when analyzing the benefits of mepolizumab in reducing surgery [43].

SNOT-22 is the most used instrument and appears to have the highest quality of developmental methodology and psychometric performance to assess patient-reported outcomes. The real improvement in terms of disease-specific HRQoL through the SNOT-22 questionnaire has been measured by only a few studies [44,47,48].

A statistically significant reduction in mean SNOT-22 scores after mepolizumab therapy compared to baseline was achieved similarly to what resulted from other studies [25,26]. The median t0 SNOT-22 score was 76 (IQR:61;90), and after treatment with mepolizumab, patients experienced a statistically significant decrease (improvement) in the t6 SNOT-22 score, with a median difference of -63 points (95% CI: −68; −58) *p* < 0.001. HRQoL also significantly improved with mepolizumab for SNOT-22 score, with a reduction of 8.9 points or higher minimal clinically significant difference.

Interestingly, the delta SNOT-22 score in our study was −63 which is almost three times that obtained from the Gallo study (−22 points) and almost five times that obtained from the SYNAPSE study (−13.7 points). This discrepancy may be possible as we present both comorbidities, patients with severe nasal polyposis and severe asthma; 89% of our patients presented uncontrolled or poorly controlled asthma as a comorbidity.

Our clinical experience with mepolizumab in patients with CRSwNP is consistent with the results of previous randomized controlled trials (RCTs) [26] and other RLS outcomes [25,47], in which patients responded favorably to mepolizumab treatment in terms of CRSwNP disease severity and HRQoL, as assessed by SNOT-22 [49]. In our study, improvement of ACT was observed in line with former post hoc meta-analysis [50].

The use of FeNO has been proposed as a marker in several respiratory diseases, including asthma [51]. Various studies showed that FeNO is frequently very high in patients with CRSwNP, so it is possible that FeNO may play a role as a biomarker typically altered in patients with nasal polyposis [52]. FeNO is performed routinely in all asthmatic patients in our series. We found a significant decreasing trend in FeNO, t0 50 (36;130) and t6 23 (14;36) *p* < 0.05, unlike Detoraki et al. [47].

Detoraki et al. [47] demonstrated a decrease in the NPS during the study, suggesting that mepolizumab treatment may reduce or delay polyp growth. As in other RLS results [45,48,53] and those reported by RCTs [26], our study showed a downward trend of NPS compared to baseline (median: t0 NPS 4; IQR:4;6) and after treatment (median: t6 NPS:1; IQR:0;1) with a median difference of −4 and a *p* < 0.001. The reduction in NPS was greater in patients with AERD. NPS score decreased by 4.5 points (95% CI: −5; −3.5) *p* < 0.001. The delta NP score was −4 points compared with that from the SYNAPSE studio (−0.8 points). We supposed it may be caused by the high incidence of AERD in our study, 51% of patients (28/55). Mepolizumab significantly reduced nasal polyp size and improved nasal obstruction symptoms in CRSwNP regardless of the presence of comorbid asthma or AERD [43].

The presence of tissue eosinophilia in CRSwNP is frequently associated with a more severe form of the disease, increased sinonasal inflammation, higher postoperative symptom scores, polyps’ recurrence and a significant impact on general and specific patient’s HRQoL [54].

Previous studies [47] have widely reported the clear and reproducible effect of mepolizumab on reducing circulating blood eosinophils.

We report a significant reduction from baseline t0 blood eosinophils 500 cell/mcl to post-treatment with mepolizumab (t6) blood eosinophils 97 cell/mcl, *p* < 0.001, representing an 80.6% reduction. In our study we found a greater reduction in the level of eosinophils among patients with AERD compared to non-AERD patients. These patients presented higher baseline blood eosinophil counts, so we assumed as in RCTs (SYNAPSE) [26] that the efficacy of mepolizumab is higher, with a significantly higher baseline blood eosinophil count. The reduction in eosinophil count was associated with a significant decrease in OCS intake from baseline to t6.

Epistaxis, headaches, nasopharyngitis and sinusitis were the most frequent adverse effects described in previous studies, and we found that the safety profile of mepolizumab was similar to those previously reported [26]. Long-term follow-up will be essential to determine the impact and frequency of these adverse effects. In patients with refractory disease and who do not respond to the standard of care treatment, the subjective and objective efficacy of mepolizumab for CRSwNP has been demonstrated in phase 3 trials, and FDA approved in July 2021 at a dose of 100 mg of administered subcutaneously [26].

If we compare mepolizumab with other biologics currently FDA approved for CRSwNP such as Omalizumab and Dupilumab, we find improvement in endoscopic, clinical and/or radiological endpoints with all of them in patients with CRSwNP with or without comorbid asthma. The choice depends on patient and provider preference and insurance coverage [55].

There are several potential limitations for this study which should be considered when interpreting the results. First, it is a retrospective analysis on a small sample size. Secondly, in Spain, mepolizumab is approved for the treatment of CRSwNP, but payment is not granted by the insurance, so except for six patients, all our patients were treated with this biologic therapy when diagnosed with asthma comorbidity. There is no control group, because all our patients were treated with mepolizumab. Radiological changes after treatment were not evaluated. In fact, we did not perform CT scans after treatment to avoid unnecessary ionizing radiation. In addition, scans are repeated only in refractory patients awaiting new surgery or in case of suspected clinical complications. The impact of surgery on the results was not evaluated given the wide variability of techniques in the type of surgical procedure (from the less aggressive endoscopic approaches such as polypectomy to the most aggressive such as the “reboot surgery” or other extensive techniques, associated or non-complementary methods such as mucoplasty, in addition to the approaches for the control of the frontal sinus as the Draft III) did not allow for comparisons between patients. Finally, asthma and AERD as comorbidities were determined from the clinical data of the patients, and the only tests performed during the study to evaluate these diseases were the ACT score and the FeNO.

## 7. Conclusions

Beneficial biological treatment effects on clinical and endoscopic aspects has been demonstrated with a significantly reduced symptoms, polyp scores, blood eosinophils and systemic corticosteroid use, resulting in an increase HRQoL in patients with severe CRSwNP, regardless of the presence or absence of comorbid asthma or comorbid AERD.

In our study treatment with mepolizumab induced a better asthma control in terms of clinical (ACT test) evaluation.

Specific mepolizumab therapy seems to be another alternative option well-tolerated for those patients with difficult-to-treat CRSwNP. Our real-life results are consistent with those from previous RCTs with mepolizumab and contribute to a better understanding of the impact of mepolizumab on symptom control, reduction in blood eosinophil count and corticosteroid use in these patients with a large healthy burden and severe disease.

Further studies with larger numbers of patients would be needed to complement these preliminary results.

## Figures and Tables

**Figure 1 biomedicines-11-00485-f001:**
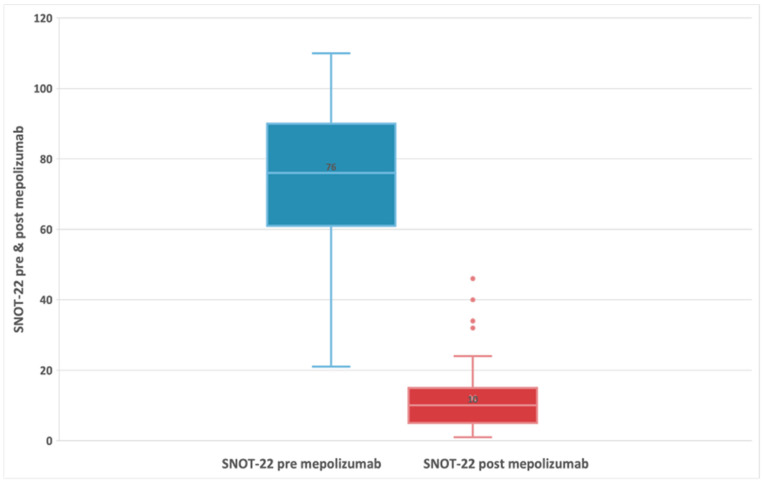
Evolution of SNOT-22 score in patients with refractory chronic rhinosinusitis and nasal polyps, pre-treatment, and 24-week post-treatment with 100 mg of mepolizumab every 4 weeks.

**Figure 2 biomedicines-11-00485-f002:**
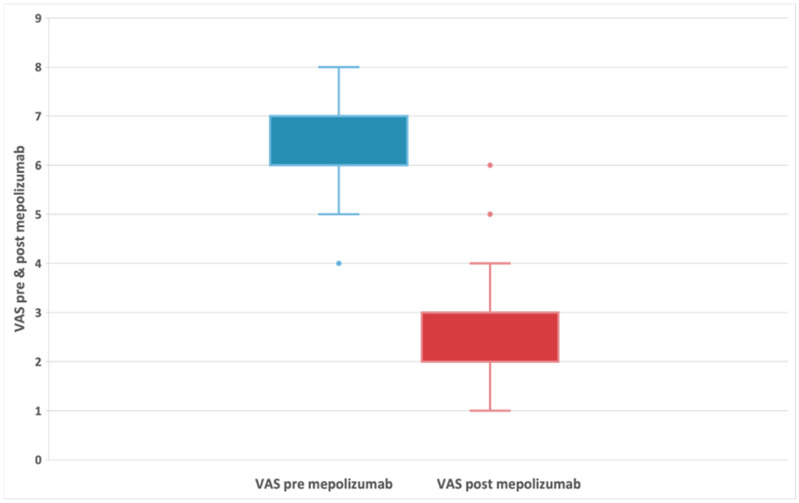
Evolution of VAS symptoms in patients with refractory chronic rhinosinusitis and nasal polyps, before and after 24-week treatment with 100 mg of mepolizumab every 4 weeks.

**Figure 3 biomedicines-11-00485-f003:**
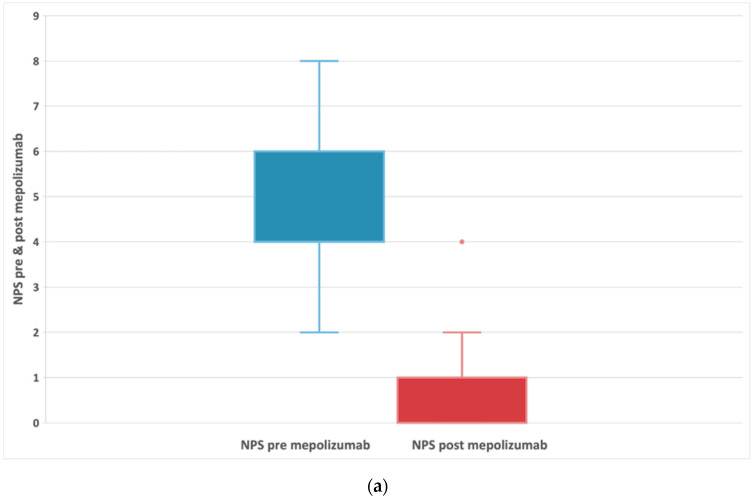
(**a**) Evolution of NPS in patients with refractory chronic rhinosinusitis and nasal polyps, pre-treatment and after 24-week treatment with 100 mg of mepolizumab every 4 weeks. (**b**) Evolution of NPS in patients with refractory chronic rhinosinusitis and nasal polyps, comparing subjects with and without AERD pre-treatment and after 24-week treatment with 100 mg of mepolizumab every 4 weeks.

**Figure 4 biomedicines-11-00485-f004:**
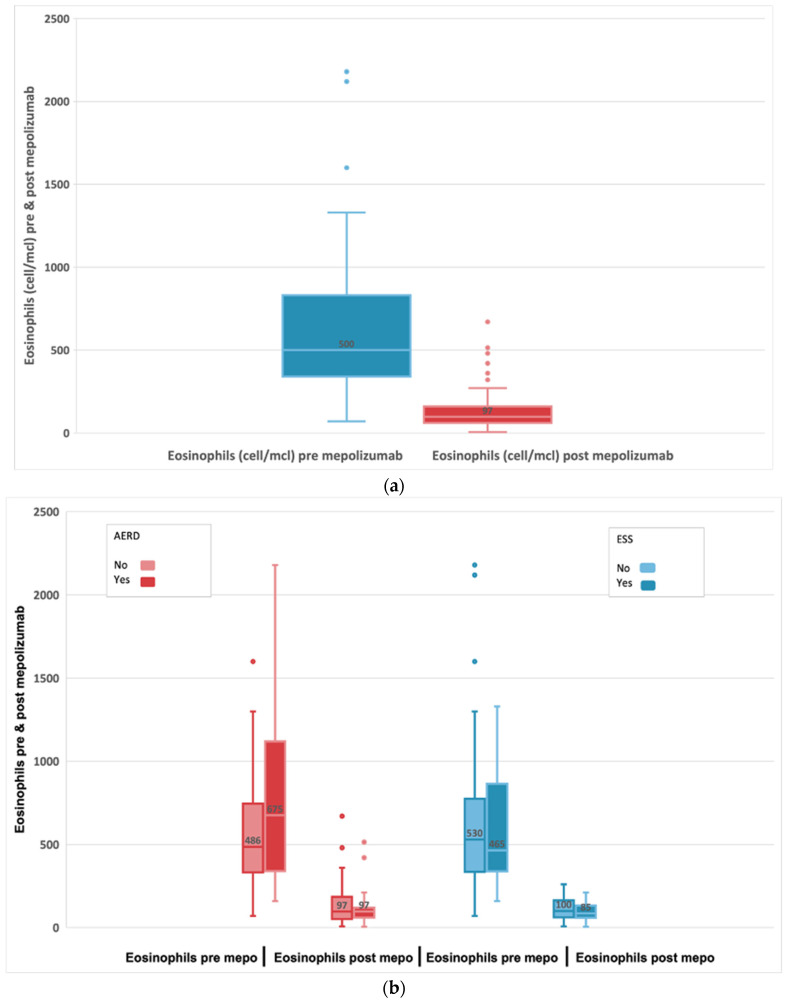
(**a**) Evolution of absolute blood eosinophils in subjects with refractory chronic rhinosinusitis and nasal polyps, pre-treatment and after 24-week treatment with 100 mg of mepolizumab every 4 weeks. (**b**) Evolution of blood eosinophils before and after 24-week treatment with 100 mg of mepolizumab every 4 weeks among patients with AERD compared to non-AERD and subjects with previous surgery compared to non-previous surgery.

**Figure 5 biomedicines-11-00485-f005:**
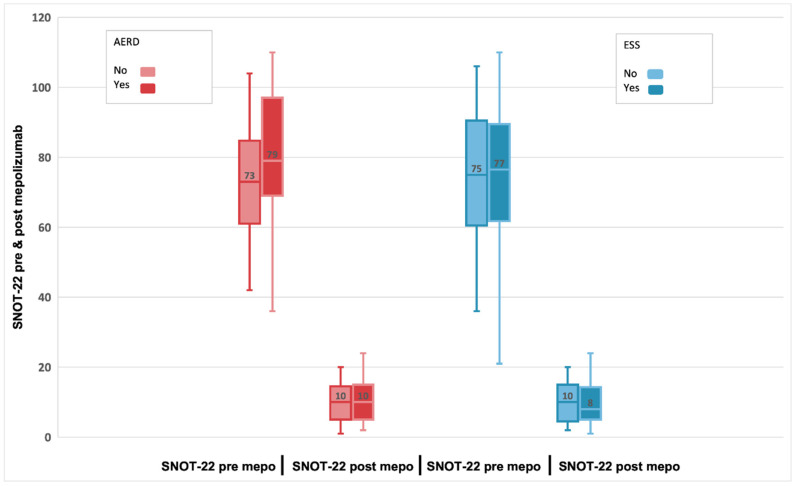
Evolution of SNOT-22 scores pre-treatment and after 24-week treatment with 100 mg of mepolizumab every 4 weeks among patients with AERD compared to non-AERD and subjects with previous surgery compared to non-previous surgery.

**Table 1 biomedicines-11-00485-t001:** Demographic and baseline clinical characteristics of the treated population.

	Male	Female	Overall
Number of patients, *n* (%)	20 (36)	35 (64)	55
Age in years, mean ± SD	53.3 ± 9.9	53.9 ± 11.7	53.7 ± 11.0
Chronic rhinosinusitis with nasal polyps (CRSwNP), yes, *n* (%)	20 (100)	35 (100)	55 (100)
Endoscopic sinus surgery (ESS), yes, *n* (%)	11 (55)	19 (54)	30 (55)
Endoscopic sinus surgery (ESS), median (IQR)	1 (0;2)	1 (0;2.8)	1 (0;2)
Asthma, yes, *n* (%)	17 (85)	32 (91)	49 (89)
Respiratory disease exacerbated by aspirin and nonsteroidal anti-inflammatory drugs (AERD), yes, *n* (%)	11 (55)	17 (49)	28 (51)
Smoker, yes, *n* (%)	5 (25)	4 (11)	9 (16)
Allergic, yes, *n* (%)	10 (50)	19 (54)	29 (53)
Corticosteroid dependent, yes, *n* (%)	15 (75)	29 (83)	44 (80)
Lund-Mackay score, median (IQR)	20 (19;2)	20 (13;23)	20 (16;22)

Data are means ± SD when normality is assumed; medians (IQR) under non-normality and frequencies (%). SD: standard deviation; IQR: interquartile range.

**Table 2 biomedicines-11-00485-t002:** Summary of patient-reported outcomes pre-Mepo = t0 and post-Mepo = t6.

	Baseline = Pre-Mepo * *n* = 55	Post-Mepo * *n* = 55	*p* Value	Median of the Differences, 95% CI
Sinonasal outcome test (SNOT-22) total score, median (IQR); range 0–110	76 (61;90)	10 (5;15)	<0.001 ^§^	−63, (−68; −58) ^†^
Eosinophils (cell/mcl), median (IQR)	500 (340;830)	97 (60;160)	<0.001 ^§^	−460, (−610; −353) ^†^
Nasal endoscopic polyp score (NPS), median (IQR); range 0–8	4 (4;6)	1 (0;1)	<0.001 ^§^	−4, (−4.5; −3.5) ^†^
Visual analogue scale (VAS), median (IQR)	6 (6;7)	2 (2;3)	<0.001 ^§^	−4, (−4; −4) ^†^
Asthma control test (ACT) total score, median (IQR); range 5–25	11 (5;25)	21 (8;25)	<0.05 ^§^	
Fractional exhaled nitric oxide (FeNO) ppb, median (IQR)	50 (36;130)	23 (14;36)	<0.05 ^§^	
Oral corticosteroid use (OCS), yes, *n* (%)	53 (96.4)	2 (3.6)	<0.001 ^‡^	
Hyposmia/Anosmia, yes, *n* (%)	52 (94.5)	19 (34.5)	<0.001 ^‡^	
Nasal Obstruction, yes, *n* (%)	52 (94.5)	14 (25.5)	<0.001 ^‡^	

* Data are means ± SD when normality is assumed; medians (IQR) under non-normality and frequencies (%). SNOT-22 scores range from 0 to 110. Lower scores imply less severe symptoms. ACT scores range from 5 to 25. Higher scores imply greater asthma control. FeNO scores range from 5 to 20 ppb. SD: standard deviation; IQR: interquartile range. ^§^ Wilcoxon signed-rank test for paired data. ^‡^ McNemar test. ^†^ Hodges–Lehmann Estimator (95% CI). The median differences between the pre-mepolizumab and post-mepolizumab scores and 95% CIs were estimated with the Hodges–Lehmann method. *p*-value is from the Wilcoxon signed-rank test.

## Data Availability

Data are available from the authors upon reasonable request.

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
