# Peer review of "Real-Life Effectiveness of Mepolizumab in Refractory Chronic Rhinosinusitis with Nasal Polyps"

_biomedicines, 2023, doi:10.3390/biomedicines11020485_

Round 1

Reviewer 1 Report

4.    1.      Interesting and actual topic, good study design

2.      Some small corrections suggested , such as : Sino-nasal, Young syndrome naso-sinusal, revision surgery might be replaced with recurrent or repeated , we found instead of we evidenced

3.      Table 1- some lines are doubled

4.      Contraindications of nasal surgeries – examples …

5.      In terms of analysis….please rephrase

6.      The safety profile – examples of the reported side effects  

7.      Comparison with other approved biologicals

8.      Comment on expected treatment  duration and disease modifying potential of biologicals 

Author Response

Report review attached

Reviewer 2 Report

The authors presented their data and results of a group of 55 patients undergoing mepolizumab in a real life scenario. The study is well set. However, I do have a few comments:

1) I believe the introduction is too long-winded about everything is already known on CRSwNP; I'd suggest to concentrate more on mAbs and what' new about them. Also, it would be interesting to read a comment on what the authors' decision was based on, to indicate mepolizumab instead of other mAbs (e.g. dupilumab).

2) It is unnecessary to detail each scale used. I believe that everybody in the field knows how the Lund-Mackay is calculated, the NPS (citation n.32 is wrong - it was not Bachert et al. to develop the NPS), the SNOT22. Their description makes the M&M unnecessarily long. 

3) Figures: if they're only box&whiskers to show the median, IQR and outliers of the outcome measures, I suggest not to put numbers on the boxes, yet report only medians.

4) Type of surgery that was performed in each patient should be better described, as not all endoscopic surgeries are the same. 

5) What is the cost of mepolizumab treatment per each patient per year and why/when is it chosen against revision surgery?

6) Another effective parameter measuring the efficacy of mAbs on upper and lower airways inflammation is exhaled nitric oxide (suggested citations to include: 

-Frendø M et al. Exhaled and nasal nitric oxide in chronic rhinosinusitis patients with nasal polyps in primary care. Rhinology. 2018 Mar 1;56(1):59-64. doi: 10.4193/Rhino17.111; 

-Paoletti G et al. J Breath Res. 2020 Nov 5;15(1):016007. doi: 10.1088/1752-7163/abc234.; 

-Paoletti G et al. Very rapid improvement of extended nitric oxide parameters, associated with clinical and functional betterment, in patients with chronic rhinosinusitis with nasal polyps (CRSwNP) treated with Dupilumab. J Investig Allergol Clin Immunol. 2022 Sep 1:0. doi: 10.18176/jiaci.0851.

-Heffler E, Pizzimenti S, Badiu I, Guida G, Ricciardolo FL, Bucca C, Rolla G. Nasal nitric oxide is a marker of poor asthma control. J Breath Res. 2013 Jun;7(2):026009. doi: 10.1088/1752-7155/7/2/026009).

I suggest the authors to make a comment on that and explain why they did not perform this measure. 

Author Response

Report review attached
